# STRUCTURED PRUNING ADAPTERS

## ABSTRACT

Adapters are a parameter-efficient alternative to fine-tuning, which augment a frozen base network to learn new tasks. Yet, the inference of the adapted model is often slower than the corresponding fine-tuned model. To improve on this, we introduce the concept of Structured Pruning Adapters (SPAs), a family of compressing, task-switching network adapters, that accelerate and specialize networks using tiny parameter sets and structured pruning. Specifically, we propose the Structured Pruning Low-rank Adapter (SPLoRA) and the Structured Pruning Residual Adapter (SPPaRA) and evaluate them on a suite of pruning methods, architectures, and image recognition benchmarks. Compared to regular structured pruning with fine-tuning, SPLoRA improves image recognition accuracy by 6.9% on average for ResNet50 while using half the parameters at 90% pruned weights. Alternatively, a SPLoRA augmented model can learn adaptations with $17\times$ fewer parameters at 70% pruning with 1.6% lower accuracy. For ViT-b/16 models, SPLoRA improves accuracy by an average of 43%-points at 75% pruned weights while learning $6.8\times$ fewer parameters. Our experimental code and Python library of adapters are available at *link-available-upon-acceptance*.

## 1  INTRODUCTION

Fine-tuning is an established approach to parameter-based transfer learning from a source model pre-trained on a large dataset to a target task with limited training data. However, the resulting model retains the same parameter count and computational characteristics as the source model, even when solving a considerably simpler task. A group of *fine-pruning* methods (Li et al. (2017); Molchanov et al. (2017); Sun et al. (2017); Yeom et al. (2021)) have combined pruning with fine-tuning to produce highly accurate compressed models which are well-suited for resource-constrained deployments such as mobile devices, robotics applications, and settings necessitating low latency.

Meanwhile, Adapters (Rebuffi et al. (2017; 2018)) have emerged as a viable alternative to fine-tuning for multi-domain deep neural networks (DNNs), where a single source DNN is specialized and sequentially used for multiple tasks. Instead of continuing training the source DNN weights directly, Adapters introduce parameter-efficient layer add-ons, which are trained instead. As these add-ons are much more compact than the source DNN weights, they can be transmitted and stored at low cost. This is very useful, *e.g.*, for edge devices and federated learning (McMahan et al. (2016)). However, prior work has largely ignored the computational efficiency aspects of Adapters, which either increase the complexity of the network (He et al. (2022); Zhu et al. (2021); Li & Liang (2021); Houlsby et al. (2019); Pfeiffer et al. (2021); Mahabadi et al. (2021)) or leave it unaltered by utilizing structures that can be fused with the original weights (Rebuffi et al. (2017; 2018); Hu et al. (2022)).

While the combination of pruning and fine-tuning can produce compressed models with good performance at an order of magnitude fewer parameters compared to the source model, we show that a Low-rank Adapter (Hu et al. (2022)) augmented with structured pruning of network channels, i.e., a *Structured Pruning Low-rank Adapter (SPLoRA)*, can improve upon this by another order of magnitude for task-switching networks. Another SPA instantiation, the *Structure Pruning Parallel Residual Adapter (SPPaRA)*, achieves similar performance on convolutional architectures. This is showcased through a comprehensive comparison to pruning with fine-tuning in weight-based transfer learning from pretrained convolutional and transformer-based networks to a battery of five image classification benchmarks, four different pruning methods, and four network architectures. Here, we find that SPAs not only reduce parameter requirements per task massively, but also retain predictive accuracy better than fine-tuning under aggressive pruning.

## 2 RELATED WORK

**Adapter methods**   When multiple specialized versions of a network are deployed on the same device and storage requirements are strict, Adapters (Rebuffi et al. (2017)) provide a low-parameter-count alternative to fine-tuning. Instead of deploying multiple sets of full network weights, a single set of full weights can be deployed alongside multiple adapter weights, which augment the main network. For Convolutional Neural Networks (CNNs), point-wise convolutions can be introduced in series (Rebuffi et al. (2017)) or parallel (Rebuffi et al. (2018)) with a residual connection to adapt fixed source weights to new tasks. For Transformer-based networks, prior work explored bottleneck projections with (Zhu et al. (2021)) and without (Hu et al. (2022)) low-dimensional non-linearity in parallel with the fixed dense layers of the original network. Adapter blocks can also be interspersed in series with existing layers (Houlsby et al. (2019); Pfeiffer et al. (2021); Mahabadi et al. (2021)). Several works (Stickland & Murray (2019); Pfeiffer et al. (2021); Rücklé et al. (2021)) explored the use of multi-task adapters. While the above-described methods succeed in learning parameter-efficient network add-ons with very small storage requirements, they often incur an additional computational cost beyond the original network. Considering that the adapted target tasks are usually simpler than the source task, it is reasonable to assume that a derived network adaptation can be learned, which reduces computational complexity as well.

**Efficiency approaches**   Multiple approaches have been proposed to reduce the compute and memory footprint of neural networks. *Knowledge distillation* (Hinton et al. (2015))utilizes a large network as a teacher for a smaller network, which has more desirable memory and computational characteristics. *Efficient architectures* (Tan & Le (2019); Feichtenhofer (2020)) define and optimize expressive yet efficient architectural blocks from random initialization under a multi-metric optimization goal. *Low-rank factorizations* (Tran et al. (2018); Guo et al. (2019)) approximate large tensor weights by factorizing them into multiple lower-rank weights. *Continual Inference Networks* (Hedegaard & Iosifidis (2022); Hedegaard et al. (2023)) reuse the network weights of prior DNNs with a temporal component and accelerate them for online stream processing via optimized computational sequences and appropriate intra-layer caching. *Quantization approaches* (Gray & Neuhoff (1998); Liang et al. (2021)) reduce model size and run-time costs via low-resolution numerical representations of network weights. Finally, *Pruning methods* entirely remove unnecessary network weight from a pre-trained model. While all of these are promising research avenues both in isolation and combination, we focus on pruning-methods hereafter.

**Pruning methods**   DNNs can be pruned at multiple levels: *Unstructured* pruning of individual weights results in sparse weight matrices, which accelerate performance on CPU but do not generally provide speedup on GPUs. On the other hand, *structured* pruning approaches, such as the pruning of entire channels (Yeom et al. (2021)) or blocks (Lagunas et al. (2021)) of networks weights, provide inference speedup across computational devices (Gray et al. (2017)). Chen et al. (2021) propose to first partition a network into "zero-invariant groups" and subsequently train a group-sparse solution.

Many studies have proposed criteria on *what* to prune. Early methods (LeCun et al. (1989); Hassibi & Stork (1992)) proposed the use of second-order Taylor expansion of the loss Hessian for weight selection. As computing the inverse of the Hessian may be computationally intractable, another approach uses a first-order Taylor approximation of the loss change due to the pruning of units (Molchanov et al. (2017)). Another work uses fast Hessian-vector products to retain low complexity (Nonnenmacher et al. (2022)). Similarly, the gradient of a weight with respect to the loss can be used for pruning selection (Sun et al. (2017)). Yeom et al. (2021) proposed an explainability-inspired approach, computing the relevance of each network component by means of Layer-wise Relevance Pruning (LRP). Among the simplest approaches is the use of weight magnitudes in pruning selection (Han et al. (2015); Li et al. (2017)). Another consideration is whether to rank and select structural units locally within a layer (keeping pruning evenly spread throughout the network) or globally, with a contest among all network layers. We utilize global selection in our experiments.

Multiple studies have also investigated *when* to prune. A simple approach is to first prune the network to the desired sparsity in one go and subsequently train the network on the target task. Another popular approach is to use an iterative pruning and fine-tuning schedule, pruning a predefined fraction of units at a time (Renda et al. (2020)). Alternatively, Automated Gradual Pruning (Zhu & Gupta (2018)) allows all weights and masking choices to be altered throughout the pruning schedule.

## 3 TRANSFER-PRUNING

Pruning is useful not only for compressing a model while retaining predictive performance, but also for transfer learning. In fact, a task can be "learned" simply by selecting the appropriate subset of weights in a network (Mallya et al. (2018); Ramanujan et al. (2020)).

Consider a large (pre-trained) source model $f_s$ and a set of $T$ target tasks for which we desire specialized target models $f_t, t \in \{1..T\}$. Under the framework of transfer learning with pruning (*transfer-pruning*), we can concurrently update and mask weights from a source model to benefit a target task $t$. Consider $g : \boldsymbol{W}_s \times \Delta \boldsymbol{W}_t \times \mathbb{M}_t \to \boldsymbol{W}_t$, a function that generates target model weights $\boldsymbol{W}_t$, given learned update weights $\Delta \boldsymbol{W}_t$, source weights $\boldsymbol{W}_s$, and a learned masking set $\mathbb{M}_t$ of retained weight indices. Given available source weights $\boldsymbol{W}_s$, every task-specific model $f_t$ can be stored as the parameters $\Phi_t = \{\Delta \boldsymbol{W}_t, \mathbb{M}_t\}$.

Under *fine-pruning* (Sanh et al. (2020)), *i.e.*, concurrent pruning and fine-tuning, $g$ constitutes a direct assignment of weights, $\boldsymbol{W}_t := g(\boldsymbol{W}_s, \Delta \boldsymbol{W}_t, \mathbb{M}_t) = \Delta \boldsymbol{W}_t$, where update weights are learned based on a pruned subset, $\{\boldsymbol{W}_s^{(i)}, i \in \mathbb{M}_t\}$. Here, the parameters of the task-specific model are $\Phi_t = \{\boldsymbol{W}_t, \mathbb{M}_t\}$, and the size of the target weights is determined by the weight density $d \in (0, 1]$ and the size of source weights, *i.e.*, $\|\boldsymbol{W}_t\|_0 = d\|\boldsymbol{W}_s\|_0$.

## 4 STRUCTURED PRUNING ADAPTERS

Although fine-pruning can successfully produce smaller target weights, the set of weights for all tasks $\{\boldsymbol{W}_t\}$ may still consume an intractable amount of storage if many tasks $T$ are involved and/or the average density $\bar{d}$ is large due to high predictive performance requirements. Instead, we seek to utilize adapters alongside pruning to produce an extremely compressed parameter set. Consider the concurrent pruning and adaptation of a frozen source projection matrix $\boldsymbol{W}_s \in \mathbb{R}^{n \times m}$ with an index mask $\boldsymbol{M} \in \{0, 1\}^{n \times m}$ and an adapter function $a$. While different applicable adapters have been extensively studied (see Section 2), we restrict ourselves to fusible parallel adapters to minimize the run-time of the resulting model. Denoting element-wise multiplication by $\odot$, Structured Pruning Adapters (SPAs) take the following basic form:

$$\boldsymbol{W}_t = (\boldsymbol{W}_s + a(\Delta \boldsymbol{W}_t)) \odot \boldsymbol{M}. \tag{1}$$

**Channel SPAs** *Channel* pruning maps a dense source matrix to a dense pruned matrix with computational improvements proportional to the number of removed parameters. A mask $\boldsymbol{M}$ in this case can be decomposed as row and column masks $\boldsymbol{m}_{\text{in}} \in \{0, 1\}^{n \times 1}$ and $\boldsymbol{m}_{\text{out}} \in \{0, 1\}^{m \times 1}$, respectively. Then, Equation (1) can be expressed as

$$\boldsymbol{W}_t = (\boldsymbol{W}_s + a(\Delta \boldsymbol{W}_t)) \odot \boldsymbol{m}_{\text{in}} \boldsymbol{m}_{\text{out}}^\top. \tag{2}$$

**Convolutional SPAs** Beyond the adaptation of linear layers, a fusible parallel adapter can also be embedded into a convolutional kernel. Consider the source kernel $\mathsf{W}_s \in \mathbb{R}^{k \times \dots \times c_i \times c_o}$, with an odd kernel size $k$ and channel dimensions $c_i$ and $c_o$. Adopting the notation for convolutional parallel residual adapters (Rebuffi et al. (2018)), where a matrix $\boldsymbol{X}$ is embedded diagonally in a zero-valued higher-dimensional tensor

$$\text{diag}_k(\boldsymbol{X})^{[x_1, \dots, i, o]} = \begin{cases} \boldsymbol{X}^{[i,o]}, & \forall_j x_j = (k-1)/2 + 1 \\ 0, & \text{otherwise}, \end{cases}$$

we can succinctly define the parallel residual adaptation of $\mathsf{W}_s$ with a $a(\Delta \boldsymbol{W}_t) \in \mathbb{R}^{c_i, \times c_o}$ under pruning with $\mathsf{M} \in \{0, 1\}^{k \times \dots \times c_i \times c_o}$ as

$$\mathsf{W}_t = (\mathsf{W}_s + \text{diag}_k(a(\Delta \boldsymbol{W}_t))) \odot \mathsf{M}. \tag{3}$$

**Structured Pruning Low-rank Adapter** A simple realization of a fusible parallel adapter is the Low-rank Adapter (LoRA) (Hu et al. (2022)):

$$\boldsymbol{W}_t = \boldsymbol{W}_s + \boldsymbol{W}_{\text{down}} \boldsymbol{W}_{\text{up}}, \tag{4}$$

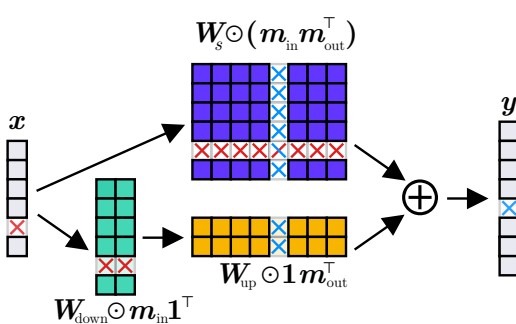

Figure 1: Structured Pruning Low-rank Adapter (SPLoRA). Pruning of in/out channels affects the adapter as well as source weights.

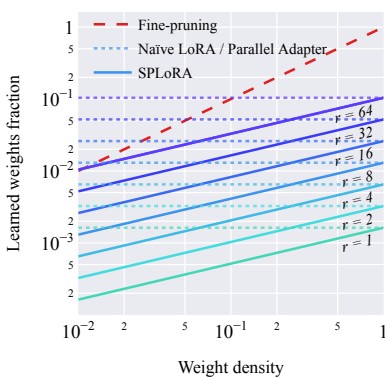

Figure 2: Learned weight fraction ($\|\Delta W_t\|_0 / \|W_s\|_0$) versus weight density ($\|W_t\|_0 / \|W_s\|_0$) for a linear layer with $768 \times 3072$ weights.

where $W_{\text{down}} \in \mathbb{R}^{n \times r}$ and $W_{\text{up}} \in \mathbb{R}^{r \times m}$ are adapter weights and $r$ is the rank hyper-parameter. Combining Equations 2 and 4 we define the **S**tructured **P**runing **Lo**w **R**ank **A**dapter (SPLoRA):

$$W_t = (W_s + W_{\text{down}} W_{\text{up}}) \odot m_{\text{in}} m_{\text{out}}^\top. \tag{5}$$

Following the derivation in Appendix A, Equation (5) can be rewritten as

$$W_t = W_s \odot m_{\text{row}} m_{\text{col}}^\top + (W_{\text{down}} \odot m_{\text{row}} \mathbf{1}^\top)(W_{\text{up}} \odot \mathbf{1} m_{\text{col}}^\top). \tag{6}$$

In this form, it is evident that channel-pruning affects not only the source weights $W_s$ in the first term, but also applies to the adapter parameters in the second term, $W_{\text{up}}$ and $W_{\text{down}}$, independently. This effect is illustrated in Figure 1.

**Convolutional Channel-Pruning Adapters**  The channel-pruning and adaptation of a linear layer can be extended to the concurrent adaptation and pruning of convolutional kernels as well. Using $a(\Delta W_t) = W_a \in \mathbb{R}^{c_i \times c_o}$ and a channel pruning mask repeated accross all kernel dimensions, $\text{rep}_k(X)^{[k_1, \cdots, i, o]} = X^{[i,o]}$, we can extend the Parallel Residual Adapter (Rebuffi et al. (2018)) as a **S**tructure **P**runing **Pa**rallel **R**esidual **A**dapter (SPPaRA):

$$\mathbf{W}_t = (\mathbf{W}_s + \text{diag}_k(\mathbf{W}_a)) \odot \text{rep}_k(m_{\text{in}} m_{\text{out}}). \tag{7}$$

Similarly, the Low-rank Adapter $a(\Delta W_t) = W_{\text{down}} W_{\text{up}}$ can be embedded in a a channel-pruned convolutional kernel to form the Convolutional SPLoRA:

$$\mathbf{W}_t = (\mathbf{W}_s + \text{diag}_k(\mathbf{W}_{\text{down}} \mathbf{W}_{\text{up}})) \odot \text{rep}_k(m_{\text{in}} m_{\text{out}}). \tag{8}$$

**Learned parameter count**  The adaptation of an $n \times m$ matrix with LoRA has $r(n + m)$ learned parameters, a fine-pruned weight has $\|m_{\text{in}}\|_0 \|m_{\text{out}}\|_0$ learned parameters, and adaptation with SPLoRA has $r(\|m_{\text{in}}\|_0 + \|m_{\text{out}}\|_0)$ learned parameters. Comparisons of learned parameter count therefore depend on both the weight density, weight matrix shape, and adapter bottleneck dimension $r$. Figure 2 visualizes the fraction of learned parameters relative to the source weight parameter as a function of weight density and $r$ for a $768 \times 3072$ matrix. SPLoRA requires fewer learned parameters than either fine-pruning or regular adapters (*e.g.*, naïve LoRA and the Parallel Adapter (He et al. (2022)).

A fine-pruned $d$-dimensional convolution with kernel size $k$ has a parameter count of $k^d \|m_{\text{in}}\|_0 \|m_{\text{out}}\|_0$ under channel pruning. The parameter count of SPPaRA as found in Equation (7) is $\|m_{\text{in}}\|_0 \|m_{\text{out}}\|_0$ and that of the convolutional SPLoRA is $r(\|m_{\text{in}}\|_0 + \|m_{\text{out}}\|_0)$.

## 5 EXPERIMENTS

We seek to compare structured pruning with fine-tuning to the use of SPAs. As both approaches have identical acceleration benefits during inference, the experimental comparison focuses on predictive performance and the number of learned parameters ($\Delta$Params).The remainder of this section compares the adaptation of convolutional layers in CNNs (Section 5.1) and linear layers in transformer-based networks (Section 5.2) using various criteria, architectures, and image classification datasets.

**Datasets** The considered models are pre-trained on ImageNet1k (Russakovsky et al. (2015)) and subsequently transfer-pruned to respectively CIFAR-10 (Krizhevsky (2009)), CIFAR-100 (Krizhevsky (2009)), Oxford Flowers 102 (Nilsback & Zisserman (2008)), Cats and Dogs (Elson et al. (2007)), or Stanford Cars (Krause et al. (2013))[1].

### 5.1 ADAPTATION OF CONVOLUTIONAL NEURAL NETWORKS

**Experiment setup** We reuse and augment a previously reported setup (Yeom et al. (2021)) to perform the transfer-pruning of convolutional channels using a filter global ranking criterion. We first train the network without pruning for $\{30, 60, 100, 100, 100\}$ epochs and subsequently prune the model at increments of 5% until 5% of weights remain in total. The pruning is interspersed with $\{20, 30, 50, 50, 50\}$ epochs of training for the {CIFAR-10, CIFAR-100, Oxford Flowers 102, Cats & Dogs, Stanford Cars} datasets. Appendix B presents an overview of training times. We employ SGD with a momentum of 0.9, weight decay of $5 \cdot 10^{-4}$, and learning rate of 0.01 at a batch size 256 or down-scaled rates following the *linear scaling rule* (Krizhevsky (2014)) when GPU memory limitations must be accomodated. In each training block, we use a step learning rate reduction of $5\times$ after each quarter of epochs. The above setup is used for either fine-pruning, in which all model weights are updated, or adaptation and pruning, which freezes the original network weights and only trains the adapter weights, normalization, and prediction head.

**SPLoRA initialization and rank choice** To gauge the sensitivity of SPLoRA hyper-parameters and their effect on predictive performance, we perform a set of adaptation and pruning runs using $L_2$-normalized Taylor pruning (Molchanov et al. (2017)) on CIFAR-10 with a ResNet-18 network. Here, we vary the rank $r \in 2^{[0,6]}$ and initialization range in $10^{[-6,-2]}$ and evaluate along densities $\{1.0, 0.5, 0.3, 0.2, 0.1\}$. As illustrated in Figure 4, we observe a clear and expected trend of increasing accuracy as the rank is increased. The increases exhibit diminishing returns and have limited benefit beyond $r = 32$ for CIFAR-10. While all tested ranks show similar accuracy at a density of $d = 1.0$, the lowest-rank adapters are more severely affected by a lower $d$. This follows intuition, considering that lower-rank adapters have fewer parameters that might prove redundant during pruning. SPLoRA is generally robust to the chosen initialization range, showing no clear trends in favor of particular ranges. We will use $\Delta W_t^{(i,j)} \sim \mathcal{U}(-10^{-4}, 10^{-4})$ in subsequent experiments.

**Effectiveness with different pruning criteria** The choice of pruning criterion can have significant impact on the quality of the resulting pruning. In this set of experiments we respectively fine-prune and adapt-and-prune a ResNet-50 (He et al. (2016)) network using four structured pruning criteria, namely, the normalized Weight (Li et al. (2017), Taylor (Molchanov et al. (2017)), Gradient (Sun et al. (2017)), and LRP (Li et al. (2017)) methods. The comparison is conducted SPLoRA with the ranks $r = 8$ and $r = 32$. To accommodate the stochastic nature of pruning and neural network training, we repeat each experiment three times and report the mean and standard deviation of each metric. The results of our experiments are presented in Table 1 for model densities of 100%, 30%, and 10% retained weights[2] and visualized for CIFAR-10 in Figure 3.

Comparing SPLoRA with fine-pruning, we observed competitive transfer adaptations on all pruning criteria, even though SPLoRA uses but a fraction the trainable weights. While fine-pruning generally resulted in higher accuracy at 30% model density (on average 0.6% and 1.6% higher than SPLoRA

---

[1]As no publicly available test split was available for Cats and Dogs, we defined train-test splits and preprocessed data using DatasetOps (Hedegaard et al. (2022)) to match the 8005 training and 2023 test samples reported previously byYeom et al. (2021). The other datasets are used with standard splits.

[2]We found LRP with fine-pruning to be unstable at low model densities for Cats & Dogs. Despite attempts with multiple different seeds, results for Cats and Dogs could not be obtained (denoted by "-" in Table 1).

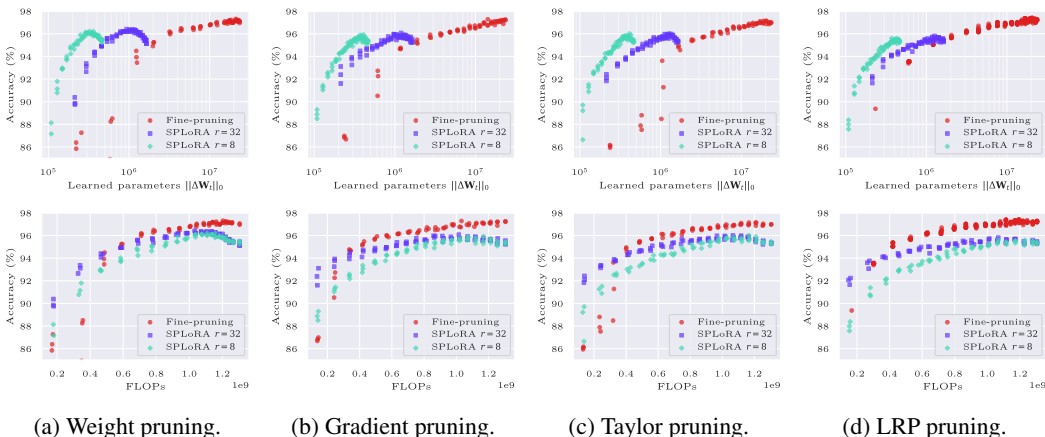

(a) Weight pruning.     (b) Gradient pruning.     (c) Taylor pruning.     (d) LRP pruning.

Figure 3: CIFAR-10 accuracy versus total learned parameter count $\|\Delta \boldsymbol{W}_t\|_0$ (top row) and FLOPs (bottom row) for (a) Weight (Li et al. (2017)), (b) Gradient (Sun et al. (2017)), (c) Taylor (Molchanov et al. (2017)), and (d) LRP (Yeom et al. (2021)) channel-pruning methods.

Table 1: Channel-based transfer-pruning from ResNet-50 pre-trained on ImageNet to Cats & Dogs, Oxford Flowers, and CIFAR-10 using Weight (Li et al. (2017)), Gradient (Sun et al. (2017)), Taylor (Molchanov et al. (2017)), and LRP (Yeom et al. (2021)) pruning. Note that SPLoRA and LoRA are identical at 100% density. $\Delta$Params and FLOPs are shown for CIFAR-10. Mean $\pm$ standard deviation is shown for each metric. Best metric per pruning-method and density is highlighted.

| Pruning method | Dens. | Learning method | $\Delta$Params (k) | FLOPs (M) | Acc. (%) CIFAR-10 | Flowers | Cats & Dogs | Avg. |
|---|---|---|---|---|---|---|---|---|
| None | 100% | Fine-tuning | $23{,}520.8_{\pm 0.0}$ | $1{,}304.7_{\pm 0.0}$ | $\mathbf{97.10}_{\pm 0.12}$ | $\mathbf{92.20}_{\pm 0.00}$ | $\mathbf{99.30}_{\pm 0.02}$ | $\mathbf{96.20}$ |
| | | LoRA-$r32$ | $1{,}644.5_{\pm 0.0}$ | $1{,}304.7_{\pm 0.0}$ | $95.32_{\pm 0.13}$ | $78.57_{\pm 5.93}$ | $98.60_{\pm 0.43}$ | $90.83$ |
| | | LoRA-$r8$ | $\mathbf{466.3}_{\pm 0.0}$ | $1{,}304.7_{\pm 0.0}$ | $95.35_{\pm 0.10}$ | $80.96_{\pm 5.83}$ | $98.84_{\pm 0.52}$ | $91.72$ |
| Weight | 30% | Fine-pruning | $4{,}427.1_{\pm 72.5}$ | $785.4_{\pm 5.4}$ | $\mathbf{96.38}_{\pm 0.12}$ | $93.64_{\pm 2.15}$ | $\mathbf{98.64}_{\pm 0.04}$ | $\mathbf{96.22}$ |
| | | SPLoRA-$r32$ | $618.3_{\pm 2.5}$ | $778.6_{\pm 2.9}$ | $95.59_{\pm 0.22}$ | $\mathbf{94.57}_{\pm 0.58}$ | $98.43_{\pm 0.13}$ | $96.20$ |
| | | SPLoRA-$r8$ | $\mathbf{210.0}_{\pm 1.9}$ | $773.4_{\pm 2.1}$ | $94.91_{\pm 0.24}$ | $92.13_{\pm 0.15}$ | $98.40_{\pm 0.12}$ | $95.15$ |
| | 10% | Fine-pruning | $599.1_{\pm 20.1}$ | $352.1_{\pm 3.9}$ | $87.23_{\pm 2.00}$ | $72.83_{\pm 0.93}$ | $95.42_{\pm 0.31}$ | $85.16$ |
| | | SPLoRA-$r32$ | $294.8_{\pm 0.4}$ | $335.4_{\pm 7.3}$ | $\mathbf{93.04}_{\pm 0.37}$ | $\mathbf{89.36}_{\pm 1.09}$ | $\mathbf{96.25}_{\pm 0.99}$ | $\mathbf{92.88}$ |
| | | SPLoRA-$r8$ | $\mathbf{128.9}_{\pm 0.1}$ | $338.9_{\pm 7.0}$ | $91.24_{\pm 0.51}$ | $86.49_{\pm 0.74}$ | $96.07_{\pm 0.69}$ | $91.27$ |
| Gradient | 30% | Fine-pruning | $3{,}719.8_{\pm 59.2}$ | $571.9_{\pm 6.5}$ | $\mathbf{95.95}_{\pm 0.09}$ | $\mathbf{94.21}_{\pm 0.74}$ | $\mathbf{98.22}_{\pm 0.00}$ | $\mathbf{96.13}$ |
| | | SPLoRA-$r32$ | $601.0_{\pm 0.4}$ | $564.6_{\pm 1.5}$ | $94.91_{\pm 0.09}$ | $93.58_{\pm 0.43}$ | $98.17_{\pm 0.08}$ | $95.55$ |
| | | SPLoRA-$r8$ | $\mathbf{205.2}_{\pm 0.2}$ | $565.2_{\pm 4.1}$ | $94.09_{\pm 0.28}$ | $91.60_{\pm 0.30}$ | $98.15_{\pm 0.07}$ | $94.61$ |
| | 10% | Fine-pruning | $615.7_{\pm 4.3}$ | $244.7_{\pm 3.3}$ | $91.83_{\pm 1.17}$ | $73.06_{\pm 0.70}$ | $95.84_{\pm 0.20}$ | $83.58$ |
| | | SPLoRA-$r32$ | $293.1_{\pm 0.4}$ | $244.0_{\pm 1.9}$ | $\mathbf{93.65}_{\pm 0.36}$ | $\mathbf{91.35}_{\pm 0.47}$ | $\mathbf{97.54}_{\pm 0.17}$ | $\mathbf{94.18}$ |
| | | SPLoRA-$r8$ | $\mathbf{128.3}_{\pm 0.0}$ | $245.4_{\pm 4.0}$ | $91.25_{\pm 0.20}$ | $87.46_{\pm 0.71}$ | $97.19_{\pm 0.28}$ | $91.97$ |
| Taylor | 30% | Fine-pruning | $3{,}392.8_{\pm 81.1}$ | $559.9_{\pm 0.7}$ | $\mathbf{95.71}_{\pm 0.02}$ | $92.91_{\pm 0.56}$ | $\mathbf{98.22}_{\pm 0.18}$ | $\mathbf{95.61}$ |
| | | SPLoRA-$r32$ | $599.7_{\pm 0.9}$ | $555.5_{\pm 6.7}$ | $94.88_{\pm 0.21}$ | $\mathbf{93.41}_{\pm 0.06}$ | $97.84_{\pm 0.48}$ | $95.37$ |
| | | SPLoRA-$r8$ | $\mathbf{205.3}_{\pm 0.1}$ | $566.2_{\pm 10.9}$ | $93.98_{\pm 0.24}$ | $91.51_{\pm 0.49}$ | $97.90_{\pm 0.13}$ | $94.46$ |
| | 10% | Fine-pruning | $576.8_{\pm 9.9}$ | $236.9_{\pm 3.3}$ | $88.07_{\pm 0.66}$ | $65.67_{\pm 4.12}$ | $95.30_{\pm 0.21}$ | $83.01$ |
| | | SPLoRA-$r32$ | $292.6_{\pm 0.5}$ | $242.0_{\pm 1.5}$ | $\mathbf{93.27}_{\pm 0.12}$ | $\mathbf{91.30}_{\pm 0.10}$ | $\mathbf{97.21}_{\pm 0.10}$ | $\mathbf{93.93}$ |
| | | SPLoRA-$r8$ | $\mathbf{128.4}_{\pm 0.1}$ | $243.2_{\pm 9.8}$ | $91.22_{\pm 0.32}$ | $86.76_{\pm 0.42}$ | $96.83_{\pm 0.30}$ | $91.60$ |
| LRP | 30% | Fine-pruning | $4{,}428.1_{\pm 20.6}$ | $719.9_{\pm 0.7}$ | $\mathbf{96.54}_{\pm 0.14}$ | $\mathbf{95.37}_{\pm 0.08}$ | $\mathbf{98.65}_{\pm 0.07}$ | $\mathbf{96.85}$ |
| | | SPLoRA-$r32$ | $592.6_{\pm 0.9}$ | $585.5_{\pm 6.7}$ | $94.85_{\pm 0.13}$ | $93.62_{\pm 0.38}$ | $98.09_{\pm 0.11}$ | $95.52$ |
| | | SPLoRA-$r8$ | $\mathbf{203.3}_{\pm 0.5}$ | $591.1_{\pm 12.4}$ | $93.53_{\pm 0.18}$ | $91.26_{\pm 0.19}$ | $97.80_{\pm 0.20}$ | $94.20$ |
| | 10% | Fine-pruning | $608.4_{\pm 6.3}$ | $301.6_{\pm 2.3}$ | $\mathbf{93.52}_{\pm 0.05}$ | $87.56_{\pm 2.75}$ | $-$ | $90.54$ |
| | | SPLoRA-$r32$ | $290.9_{\pm 0.2}$ | $270.4_{\pm 6.4}$ | $93.47_{\pm 0.36}$ | $\mathbf{91.22}_{\pm 0.46}$ | $\mathbf{97.01}_{\pm 0.27}$ | $\mathbf{93.90}$ |
| | | SPLoRA-$r8$ | $\mathbf{128.0}_{\pm 0.1}$ | $281.6_{\pm 1.7}$ | $90.94_{\pm 0.39}$ | $85.69_{\pm 1.39}$ | $96.88_{\pm 0.52}$ | $91.17$ |

ranks 32 and 8), SPLoRA had far fewer learned parameters (on average $6.2\times$ and $17.0\times$). At 10% density, SPLoRA was both more robust to pruning (achieving $6.9\%$ and $4.7\%$ higher average accuracy than fine-pruning for ranks 32 and 8), while reducing the number of learned parameters by

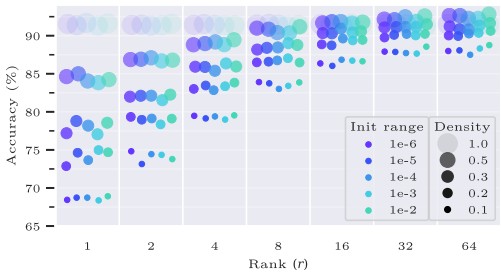
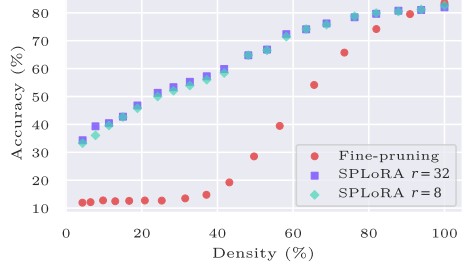

Figure 4: ResNet-18 accuracy on CIFAR-10 using SPLoRA of varying ranks ($r$) and adapter weight initialization ranges at different model densities ($d$). Point size $\propto d$ with transparency $\propto -d$.

Figure 5: Accuracy/density tradeoff for transfer-pruned ViT-b/16 on CIFAR-100.

$2.0\times$ and $4.2\times$ on average. As floating point operations (FLOPs) follow model densities, these are approximately equal for each learning method given equal densities.

**Generalization to different architectures**    We evaluate the learning methods fine-pruning, convolutional SPLoRA (Equation (8)), and SPPaRA (Equation (7)) with Gradient pruning (Sun et al. (2017)) on the VGG-16 (Simonyan & Zisserman (2015)), ResNet-50, and EfficientNetV2-M (Tan & Le (2021)) architectures. We utilize the same hyper-parameters a those reported in the respective works, but scale learning rates linearly to accommodate batch size constrains. Given the low standard deviations observed in Table 1, we only run each experiment configuration with one seed.

As presented in Table 2, we observe that fine-pruning achieves higher average accuracy than (SP)LoRA and (SP)PaRA on the ResNet-50 and EfficientNetV2-M architectures at high (100%) and medium (30%) model density. This performance gap is closed and reversed at low density (10%) where the SPAs generally outperform fine-pruning. For VGG-16, fine-pruning performed worse than either of the SPAs on all model densities[3]. Learned parameter count was significantly lower for the SPAs than for fine-pruning in all cases. Comparing SPPaRA, SPLoRA-$r32$, and SPLoRA-$r8$, the predictive accuracy and parameter-count follow a common trend: SPLoRA-$r32$ has most parameters and achieves the highest accuracy, SPPaRA has fewer parameters and scores slightly worse, and SPLoRA-$r8$ has fewest parameters, and scores the lowest average accuracy of the three SPAs. The choice of learning method should thus depend on the desired parameter count.

**Do adapters retain knowledge better than fine-tuning during pruning?**    Our experimental results on varying pruning criteria and CNN architectures show that SPAs have a somewhat surprising robustness to pruning compared with fine-tuning at low model densitites. To offer an explanation of why this phenomenon occurs, consider the following: An adapter is fundamentally anchored to the source weights and will slowly revert to the initial position due to the weight decay loss term if another learning signal does not prevent it from doing so. Even if an aggressive pruning iteration disrupts or invalidates the learned adaptations from a prior iteration the adapted weights will remain achored to the source weights. Regular fine-tuning, does not have this capability and once the model weights are fitted to a particular sparsity pattern, they cannot revert to the outset. This was experimentally corroborated by a comparison iterative pruning with one-shot pruning in Appendix C.

### 5.2 ADAPTATION OF VISION TRANSFORMERS

This section offers an experimental comparison of fine-pruning and SPLoRA for the linear layers of a Vision Transformer (ViT) (Dosovitskiy et al. (2021)). Here, a ViT-b/16 model pre-trained on ImageNet1k is transfer pruned to {Oxford Flowers 102 , Cats & Dogs, CIFAR-100, and Stanfard Cars} using {20, 50, 50, 50} initial epochs followed by iterative pruning steps with {10, 20, 20, 20} training epochs. A one-cycle learning rate scheduler with 30% warm-up and cosine annealing

---

[3]A significant portion of VGG-16 parameters reside in linear layers, which were not pruned in these experiments. While the parameter savings of using a convolutional SPA was not large for VGG-16, a linear SPLoRA can be used to reduce the learned parameter count substantially.

Table 2: Channel-based transfer-pruning from convolutional architectures, VGG-16, ResNet-50, and EfficientNetV2-M, pretrained on ImageNet1k to CIFAR-10, OxfordFlowers 102, Cats & Dogs, CIFAR-100 and Stanford Cars using Gradient pruning (Sun et al. (2017)). Learned parameters ($\Delta$Par.) and floating point operations (FLOPs) are shown for CIFAR-100. Best metric per pruning-method and density is highlighted. Note that VGG-16 contains 119,955k linear $\Delta$Par. which are not pruned. Moreover, SPLoRA/LoRA and PaRa/SPPaRA, respectively, are identical at 100% density.

| Arch. | Dens. | Learning method | $\Delta$Par. (k) | FLOPs (M) | Acc. (%) | | | | | |
|---|---|---|---|---|---|---|---|---|---|---|
| | | | | | C10 | Flwrs | C&D | C100 | Cars | Avg. |
| VGG-16 | 100% | Fine-tuning | 134,670 | 434 | 89.08 | 87.90 | 98.71 | 69.56 | 58.33 | 80.72 |
| | | PaRA | 134,797 | 434 | 91.85 | 90.62 | 98.96 | 74.77 | 67.23 | 84.69 |
| | | LoRA-$r32$ | 120,234 | 434 | **91.87** | 90.51 | **99.01** | 75.17 | **67.29** | **84.78** |
| | | LoRA-$r8$ | **120,054** | 434 | 91.84 | 90.56 | 98.96 | 74.63 | 66.98 | 84.59 |
| | 30% | Fine-pruning | 121,106 | 248 | 58.55 | 37.09 | 81.94 | 29.02 | 7.94 | 42.91 |
| | | SPPaRA | 119,994 | 238 | 89.79 | 77.85 | 95.19 | 72.17 | 68.87 | 80.77 |
| | | SPLoRA-$r32$ | 120,032 | 231 | **90.33** | **78.29** | 95.73 | 72.75 | **70.51** | **81.52** |
| | | SPLoRA-$r8$ | **119,975** | 228 | 89.39 | 77.69 | **96.63** | 69.64 | 65.85 | 79.84 |
| | 10% | Fine-pruning | 120,075 | 170 | 16.63 | 3.11 | 48.02 | 2.10 | 0.85 | 14.14 |
| | | SPPaRA | 119,967 | 165 | 88.45 | 70.26 | 92.61 | 70.13 | 64.40 | 77.17 |
| | | SPLoRA-$r32$ | 119,981 | 164 | **88.49** | **71.26** | 92.91 | 71.46 | **66.90** | **78.20** |
| | | SPLoRA-$r8$ | **119,962** | 162 | 86.30 | 69.81 | **93.11** | 66.62 | 57.40 | 74.65 |
| ResNet-50 | 100% | Fine-tuning | 23,705 | 1,305 | **97.10** | **92.20** | **99.30** | **84.22** | **88.91** | **92.35** |
| | | PaRA | 1,191 | 1,305 | 93.52 | 80.65 | 98.81 | 79.31 | 75.31 | 85.52 |
| | | LoRA-$r32$ | 1,824 | 1,305 | 95.32 | 78.57 | 98.60 | 79.47 | 87.24 | 87.84 |
| | | LoRA-$r8$ | **632** | 1,305 | 95.35 | 80.96 | 98.84 | 79.22 | 86.22 | 88.12 |
| | 30% | Fine-pruning | 3,923 | 548 | **95.95** | **94.21** | **98.22** | **79.24** | **91.70** | **91.86** |
| | | SPPaRA | 522 | 549 | 92.73 | 93.80 | 98.02 | 77.17 | 88.96 | 90.14 |
| | | SPLoRA-$r32$ | 784 | 552 | 94.91 | 93.58 | 98.17 | 78.53 | 90.05 | 91.05 |
| | | SPLoRA-$r8$ | **389** | 543 | 94.09 | 91.60 | 98.15 | 73.92 | 86.02 | 88.76 |
| | 10% | Fine-pruning | 814 | 235 | 91.83 | 73.06 | 95.84 | 67.60 | 76.94 | 81.05 |
| | | SPPaRA | 367 | 234 | 90.52 | 90.67 | **97.87** | 71.72 | 86.46 | 87.45 |
| | | SPLoRA-$r32$ | 477 | 239 | **93.65** | **91.35** | 97.54 | **75.14** | **88.72** | **89.28** |
| | | SPLoRA-$r8$ | **312** | 230 | 91.25 | 87.46 | 97.19 | 66.31 | 80.31 | 84.50 |
| EffNetV2-M | 100% | Fine-tuning | 992,450 | 120 | **98.37** | **97.54** | 99.40 | 88.33 | **91.69** | **95.07** |
| | | PaRA | 8,440 | 120 | 97.28 | 94.19 | **99.45** | 86.24 | 85.31 | 92.49 |
| | | LoRA-$r32$ | 15,061 | 120 | 96.96 | 95.10 | 99.36 | **86.35** | 85.06 | 92.57 |
| | | LoRA-$r8$ | **5,450** | 120 | 96.92 | 94.71 | **99.45** | 85.97 | 85.10 | 92.43 |
| | 30% | Fine-pruning | 98,600 | 73 | **96.75** | **97.04** | **99.16** | **84.36** | **92.16** | **93.89** |
| | | SPPaRA | 2,511 | 69 | 93.83 | 95.85 | 98.31 | 78.15 | 90.57 | 91.34 |
| | | SPLoRA-$r32$ | 4,566 | 69 | 93.22 | 96.60 | 98.31 | 79.19 | 90.29 | 91.52 |
| | | SPLoRA-$r8$ | **1,489** | 68 | 92.88 | 95.51 | 98.51 | 77.07 | 89.03 | 90.60 |
| | 10% | Fine-pruning | 7,984 | 43 | **95.35** | 80.49 | **97.62** | **82.53** | 86.07 | 88.41 |
| | | SPPaRA | 1,156 | 41 | 92.24 | 90.89 | 97.02 | 73.84 | 86.55 | 88.11 |
| | | SPLoRA-$r32$ | 1,846 | 42 | 92.06 | **92.66** | 97.17 | 76.13 | **88.37** | **89.28** |
| | | SPLoRA-$r8$ | **812** | 41 | 90.91 | 89.88 | 97.57 | 71.31 | 83.59 | 86.65 |

is used for each training block and learning rates were selected based a grid search from $10^{-4}$ to 10. For fine-pruning, we utilize a maximum learning rate of 0.0125 during the initial training and a maximum learning rate of 0.00125 after pruning at a batch size of 32. For SPLoRA, we found higher maximum learning rates of 0.375 and 0.125 to be beneficial. This discrepancy can be partly explained by the adapter initialization, which has a similar effect to changes in learning rate (Hu et al. (2022)), and partly by the following intuition: During fine-tuning, the learning process should gently nudge existing weights and avoid catastrophic forgetting of prior knowledge. Adapters, on the other hand, are anchored around the source weights and can more easily "regain knowledge", should the adaptation go slightly astray.

To perform our pruning, we utilize the Torch Pruning Python library (Fang et al. (2023)) and a weight magnitude pruner with consistent structural sparsity and channel rounding to the number attention heads. Initial runs using multiple seeds showed accuracy standard deviations of less than 0.1%-points. Given the low variance, the reported runs are conducted with only a single seed.

Table 3: Channel-based transfer-pruning from ViT-b/16 pre-trained on ImageNet1k to OxfordFlowers 102, Cats & Dogs, CIFAR-100 and Stanford Cars using weight magnitude pruning. Learned parameters ($\Delta$Par.) and floating point operations (FLOPs) are shown for CIFAR-100. Best metric per density is highlighted with bold.

| Density | FLOPs (G) | Learning method | $\Delta$Par. (k) | Acc. (%) | | | | |
|---|---|---|---|---|---|---|---|---|
| | | | | **Flwrs** | **C&D** | **C100** | **Cars** | **Avg.** |
| 100% | 17.6 | Fine-tuning | 85,286 | **97.00** | 99.46 | **83.59** | 85.98 | **91.51** |
| | | LoRA-$r32$ | 5,659 | 95.76 | 99.31 | 81.97 | **86.78** | 90.96 |
| | | LoRA-$r8$ | **1,678** | 95.37 | **99.51** | 82.64 | 86.67 | 91.05 |
| 48% | 8.9 | Fine-pruning | 42,237 | 89.63 | 94.20 | 28.53 | 48.94 | 65.33 |
| | | SPLoRA-$r32$ | 4,232 | 90.96 | **97.83** | 64.77 | **81.72** | **83.82** |
| | | SPLoRA-$r8$ | **1,218** | **91.22** | 97.33 | **64.82** | 80.01 | 83.35 |
| 24% | 4.6 | Fine-pruning | 21,409 | 44.68 | 64.71 | 12.68 | 4.72 | 31.70 |
| | | SPLoRA-$r32$ | 3,416 | 82.70 | 93.43 | **51.32** | 69.98 | **74.36** |
| | | SPLoRA-$r8$ | **955** | **82.75** | **93.97** | 50.01 | 41.28 | 67.00 |

The results in Table 3 shows similar trends as the adaptation of CNNs: Prior to pruning, simple fine-tuning slightly outperforms LoRA. However, after a few pruning steps, the accuracy drop for fine-pruning outpaces that of SPLoRA. At 24% density, the fine-pruning accuracy is severely diminished to 31.70% average accuracy, while the SPLoRA-$r32$ configuration achieves 74.36% average accuracy; 42.66%-points higher while using $6.3\times$ fewer learned weights. A detailed accuracy/density trade-off for the CIFAR-100 dataset is illustrated in Figure 5 and shows the evident discrepancy in capability between fine-pruning and SPLoRA under iterative pruning.

## 5.3 LIMITATIONS AND FUTURE WORK

**Predictive superiority may depend on the pruning approach** While this work has covered a multitude of pruning criteria and architectures on a suite of datasets, the experiments were nearly all conducted using iterative pruning. Only a few training runs used other approaches (see Appendix C) and further work is thus requires to state predictive superiority in general.

**Deployment format** For $T$ models deployed on one device, the adapter format saves space when $T > 1/\overline{d}$, assuming that available source weights, $\boldsymbol{W}_s$, are available as well. The superiority of the adapter storage format thus depends on the average density $\overline{d}$, with more aggressive pruning requiring more deployed models for SPA formats to consume less storage space.

**Pruning of adapter-specific parameters** While this work has focused on pruning of the fused weights, $\boldsymbol{W}_t$, the adapters could also be subject to pruning themselves, as explored by Rücklé et al. (2021). For instance, the SPLoRA bottleneck channels could pruned. While such intra-adapter pruning does not speed up inference, it could be an interesting avenue of future work to explore its impact on parameter count and accuracy.

## 6 CONCLUSION

We proposed Structured Pruning Adapters (SPAs) as an alternative to fine-tuning during structured pruning. Instead of updating all model weights, SPAs consist of prunable lightweight add-on modules, which are learned in place of the original weights but can be fused with them at run-time to obtain the same computational enhancements as regular structured pruning with fine-tuning. Our channel-based SPAs were shown to achieve competitive performance across a battery of pruning methods and architectures on computer vision benchmarks while requiring a fraction of the learned parameters per task. For highly pruned models, SPAs significantly outperform fine-pruning. Thus, SPAs are ideal for task-switching storage-constrained and/or network-limited usage scenarios, where the per-model size should be small, as well as the adaptation of networks under aggressive pruning.

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

## A  SPLoRA Derivation

Utilizing Equation (4) in the context of channel-SPAs of the form expressed in Equation (2), we can derive the Structured Pruning Low-rank Adapter defined in Equation (6), where adapter parameters are pruned alongside source weights. This derivation is straightforward, considering that the application of a structured pruning mask $M = m_{\text{in}} m_{\text{out}}^\top$ via Hadamard products is equivalent to a projection with diagonalized masking vectors:

$$W \odot m_{\text{in}} m_{\text{out}}^\top = \text{diag}(m_{\text{in}})\, W\, \text{diag}(m_{\text{out}}). \tag{9}$$

Similarly, a single diagonalized mask can be expressed via Hadamark products:

$$\text{diag}(m_{\text{in}})\, W = W \odot m_{\text{in}} 1^\top. \tag{10}$$

Utilizing Equation (4), Equation (9), and Equation (10), we can rewrite Equation (2) as follows:

$$\begin{aligned}
W_t &= (W_s + a(\Delta W_t)) \odot m_{\text{in}} m_{\text{out}}^\top \\
&= (W_s + W_{\text{down}} W_{\text{up}}) \odot m_{\text{in}} m_{\text{out}}^\top \\
&= \text{diag}(m_{\text{in}})\, (W_s + W_{\text{down}} W_{\text{up}})\, \text{diag}(m_{\text{out}}) \\
&= \text{diag}(m_{\text{in}})\, W_s \text{diag}(m_{\text{out}}) \\
&\qquad + (\text{diag}(m_{\text{in}}) W_{\text{down}})(W_{\text{up}} \text{diag}(m_{\text{out}})) \\
&= W_s \odot m_{\text{in}} m_{\text{out}}^\top + (W_{\text{down}} \odot m_{\text{in}} 1^\top)(W_{\text{up}} \odot 1 m_{\text{out}}^\top),
\end{aligned}$$

where the final result is equivalent to Equation (6).

## B  Training Durations

In this section, we provide a brief overview of approximate training durations for the methods tested in the present paper. As training times are comparable among different pruning methods, we report a single metric approximated from multiple pruning methods. These are presented in Table 4 for our experiments using ResNet-50 in image recognition tasks. Here, the pruning methods gradually reduce the network density while producing pruned models at a predefined step reduction in density, cycling the learning rate for each density reduction step. Accordingly, the noted training times for the pruned learning methods in Table 4 includes the training of all models with densities ranging from 100% to 5% at 5% intervals.

Table 4: Training durations for the ResNet-50 model on image recognition transfer tasks using a NVIDIA RTX 2080 Ti GPU. For each dataset, the batch size (BS) and training duration (T) are presented.

| Pruning method | Learning method | CIFAR-10 | | Flowers | | C&D | |
|---|---|---|---|---|---|---|---|
| | | BS | T | BS | T | BS | T |
| Unpruned | Fine-tuning | 64 | 0:30h | 64 | 1:05h | 64 | 2:20h |
| | SPLoRA-$r32$ | 64 | 0:30h | 32 | 1:10h | 32 | 2:30h |
| | SPLoRA-$r8$ | 64 | 0:30h | 32 | 1:10h | 32 | 2:30h |
| Pruned | Fine-pruning | 64 | 10h | 4 | 25h | 6 | 42h |
| | SPLoRA-$r32$ | 64 | 11h | 4 | 33h | 6 | 48h |
| | SPLoRA-$r8$ | 64 | 11h | 4 | 31h | 6 | 45h |

## C  SPAs RETAIN PERFORMANCE BETTER THAN FINE-PRUNING UNDER REPEATED RE-TRAINING

To offer experimental support of the hypothesis that SPAs retain performance better than fine-pruning under repeated re-training, we perform a set of experiments comparing iterative pruning, *i.e.*, the gradual pruning of 5% of model weights followed by model training at each iteration, with one-shot pruning, *i.e.*, pruning of all structural units in one go to hit the target density, in Appendix C. Here, the models were trained for 40 epochs initially and for 20 epochs after each pruning iteration under iterative pruning or for 40 epochs under one-shot pruning. The reader should bear in mind that the model accuracy is significantly reduced right after pruning in either case (an illustration of the accuracy over a complete training run is shown in Figure 7 for an example). This shows the non-trivial impact of weight removal, even for a small weight-fraction at a time.

Figure 6 shows the relative model accuracy of a ResNet-18 model pretrained on ImageNet1k and transfer-pruned to CIFAR-10 using Weight pruning (Li et al. (2017)) with both approaches. For fine-pruning, the one-shot approach performs better than iterative approach, while the iterative setting works better than the one-shot setup for SPLoRA. This is in line with the hypothesis that repeated retraining can lead to catastrophic forgetting of the beneficial source knowledge for fine-pruning. SPAs, on the other hand, do not suffer catastrophic forgetting with respect to source weights and benefit from the additional training. This finding also suggests that prior methods using fine-tuning with pruning (Molchanov et al. (2017); Liu & Wu (2019); Yeom et al. (2021)) might benefit from either switching to one-shot pruning or using a Structured Pruning Adapter.

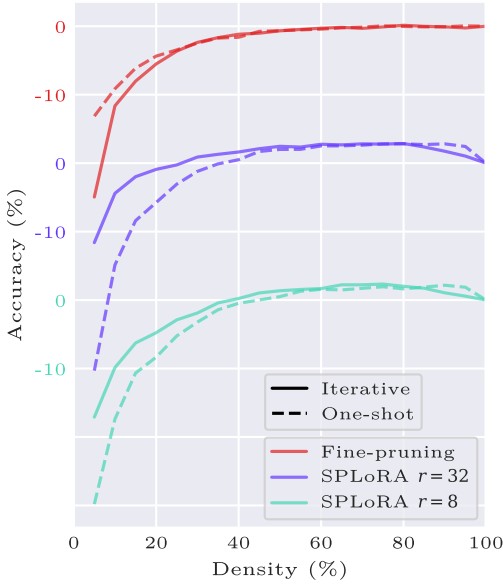

Figure 6: Comparison of one-shot and iterative transfer-pruning of a ResNet-18 pre-trained on ImageNet1k to CIFAR-10. The vertical axis denotes the accuracy relative to each learning method at 100% density.

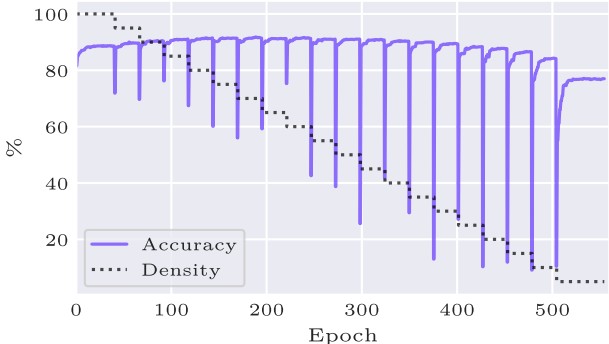

Figure 7: Iterative transfer pruning of a ResNet-18 pre-trained on ImageNet1k to CIFAR-10 using SPLoRA-$r32$. Accuracy and model density are noted as percentages.

