# OpenReview forum: "Structured Pruning Adapters"
_ICLR.cc/2024/Conference — ICLR 2024 Conference Withdrawn Submission_

### Official Review · Reviewer_onCc · 2023-10-27

**Soundness:** 3 good
**Presentation:** 4 excellent
**Contribution:** 1 poor
**Rating:** 1
**Confidence:** 4

**Summary:**

The authors propose to combine low rank foldable adapters and structured pruning encoded with a masking tensor. The resulting method can be applied to both linear and conv layers. Stemming from the success of low rank adapters and the ability of pruning to enable efficient inference, the proposed method reaches a new state-of-the-art in terms of low cost fine-tuning for efficient inference.

**Strengths:**

I think this paper is very well written. It leverages trending methods in the field, which shows the knowledge of the authors in the matter. The results are clearly superior to relevant baselines.

**Weaknesses:**

My main concern is the lack of novelty. The authors do not innovate w.r.t. pruning nor low rank adapters. Furthermore, the proposed combination of the two is straightforward. In my opinion, a naive combination of two well known method does not constitute a scientific contribution.

I would advise the author to either highlight their specific contribution to the domain of pruning or adapters, if there are any. Second, if there are no contributions to those fields, I would recommend exposing how the proposed combination is non-trivial, e.g. by providing relevant alternative combination examples (or baselines) of pruning and adapters.

**Questions:**

As stated in the weaknesses, in order to raise my rating, I need the author to refute my conclusion that the presented article does not offer any contributions to the domain of pruning nor adapters and only studies a naive combination of the two.

---

### Official Review · Reviewer_JLMs · 2023-10-29

**Soundness:** 2 fair
**Presentation:** 2 fair
**Contribution:** 2 fair
**Rating:** 3
**Confidence:** 5

**Summary:**

This paper introduces structured pruning adapters, which use lightweight prunable add-on modules instead of updating all model weights during the fine-tuning based structured pruning. It conducts evaluation on a suite of pruning methods, architectures, and image recognition benchmarks, demonstrating that the proposed method not only reduces parameter requirements per task, but also retains accuracy better than fine-tuning under aggressive pruning.

**Strengths:**

1. Extensive experiments, including 5 image classification tasks, four different pruning methods, and four model architectures.
2. The proposed method can reduce the trainable parameters during pruning.

**Weaknesses:**

While the paper is easy to read and conducts extensive experiments to demonstrate the effectiveness of introducing adapters during structured pruning, it faces several key limitations:
1. It lacks novelty. The concept of introducing LoRA adapters to reduce the number of trainable parameters is intuitive and straightforward.
2. The paper does not delve into any technical challenges, possibly due to the straightforward nature of the idea.
3. Overall, the work reads more like an engineering report.

**Questions:**

In the experiments, structured pruning adapters outperform fine-pruning at high sparsity. Can the authors provide some explanations on this?

---

### Official Review · Reviewer_dPLA · 2023-11-03

**Soundness:** 2 fair
**Presentation:** 3 good
**Contribution:** 3 good
**Rating:** 5
**Confidence:** 4

**Summary:**

This paper proposes Structured Pruning Adapters (SPAs) for accelerating and specializing net- works using tiny parameter sets and structured pruning. The evaluation shows that SPAs outperform fine-tuning for transfer pruning.

**Strengths:**

1. The writing is clear.
2. In the evaluation, it’s good that the authors repeat each experiment three times and report the mean and standard deviation of each metric.

**Weaknesses:**

1. The discussion of related works is not sufficient. The discussion about transfer pruning works is missing.
2. In Section 2, it would be better if the authors discuss the difference between the proposed method and related works, the limitations of the related methods, and which pruning category (e.g. iterative pruning v.s. one-shot pruning, structured pruning v.s. unstructured pruning) the proposed method belongs to. Although we can tell which pruning category SPAs belong to in the later sections, it can be more clear if the authors point it out.
3. Novelty is limited. The proposed method applies structured pruning in the domain of transfer learning, while these two domains are well studied.
4. Baselines are restricted. Only fine-tuning is considered as the baseline. The SOTA transfer pruning works should also be compared.
5. Most of the evaluation focuses on the ResNet and VGG architectures. It will be better if more model architectures are evaluated, like DenseNet, MobileNet, and ShuffleNet.

**Questions:**

1. Besides fine-tuning, can the authors compare SPAs with other baselines in the domain of transfer pruning?
2. In “SPLoRA initialization and rank choice”, the authors vary the rank r ∈ 2^[0,6] and initialization range in 10^[−6,−2] and evaluate along densities {1.0, 0.5, 0.3, 0.2, 0.1}. Is there a rationale for the selection of these numbers? Can the authors explain more? For example, why are densities set to {1.0, 0.5, 0.3, 0.2, 0.1}? How about other numbers, like 0.4, 0.6, 0.7, 2.0, 3.0, etc?

---

### Official Review · Reviewer_Nsqg · 2023-11-08

**Soundness:** 3 good
**Presentation:** 3 good
**Contribution:** 3 good
**Rating:** 6
**Confidence:** 4

**Summary:**

The papers contributions of this paper are the introduction of Structured Pruning Low-rank Adapter (SPLoRA) and Structure Pruning Parallel Residual Adapter (SPPaRA). A method for pruning adapters which have become a popular alternatively to full fine-tuning.

The conduct extensive experiments detailing the trade offs of pruning SPLoRA as compared to full-finetuning models on multiple datasets and model architectures. They find ins some circumstances their structured pruning adapters are competitive or outperform fine-pruning on accuracy.

**Strengths:**

the proposed technique of SPLoRA enables interesting combination of applications allowing practitioners to deploy a model fine-tuned for a variety of tasks while also accelerating them in a way thats very flexible.

They demonstrate that SPLoRA can be more effective than fine-pruning for more extreme levels of sparsity, which may be important for mobile and edge applications.

Experiments are extensive and convincing.

Paper is written in a clear and direct way.

**Weaknesses:**

Its a bit difficult to understand the utility here. Looking at the results in figure 3 and table 1 it seems that SPLoRA is strictly worse than pruning and finetuning until one reaches extreme levels of sparsity. Even then the drop in accuracy (for example at 10% density) is significant enough that it may be unacceptable.

I would also like to note it was a bit hard to understand exactly what experiments are being conducted here. I would have appreciated more self encapsulation of the experimental setup then relying on mentioned replicating another papers experimental setup.

The strongest results seem to be for the ViT transformer model where for nearly all model densities SPLoRA is significantly better than fine-tuning. A stark contrast to convolutional networks. This however begs the question why was this not explored on any Language models with transformer models are far more popular?

**Questions:**

The experimental section describes the experiments conducted as  follows:

We reuse and augment a previously reported setup (Yeom et al. (2021)) to
perform the transfer-pruning of convolutional channels using a filter global ranking criterion. We
first train the network without pruning for {30, 60, 100, 100, 100} epochs and subsequently prune
the model at increments of 5% until 5% of weights remain in total. The pruning is interspersed with
{20, 30, 50, 50, 50} epochs of training for the {CIFAR-10, CIFAR-100, Oxford Flowers 102, Cats
& Dogs, Stanford Cars} datasets.


Can you explain the decision behind why the number of pre-training epochs and fine-tuning epochs where chosen? These seem very arbitrary. Would it not have been more informative to have used the same base pre-trained model weights for all fine-tuning experiments instead of training to different numbers?


This method seems to suggest that one would like to deploy a backbone model for a variety of tasks and SPLoRA will allow you to accelerate that model as well as adapt the tasks while preserving the size on disk. Tuning a model for acceleration in deployment is a very difficult specific tasks. Is it really likely that people would like to deploy *multiple* accelerated models based on the same pre-trained model? If they have strict criteria for deployment would they not be willing to fine-tune models directly?